# Flutter to tumble transition of buoyant spheres triggered by rotational inertia changes

Varghese Mathai[1], Xiaojue Zhu [1], Chao Sun [1,2] & Detlef Lohse[1,2,3]

Heavy particles sink straight in water, while buoyant bubbles and spheres may zigzag or spiral as they rise. The precise conditions that trigger such path-instabilities are still not completely understood. For a buoyant rising sphere, two parameters are believed to govern the development of unsteady dynamics: the particle's density relative to the fluid, and its Galileo number. Consequently, with these parameters specified, the opportunities for variation in particle dynamics appear limited. In contrast to this picture, here we demonstrate that vigorous path-oscillations can be triggered by modulating a spherical particle's moment of inertia (MoI). For a buoyant sphere rising in a turbulent flow, MoI reduction triggers a tumble–flutter transition, while in quiescent liquid, it induces a modification of the sphere wake resulting in large-amplitude path-oscillations. The present finding opens the door for control of particle path- and wake-instabilities, with potential for enhanced mixing and heat transfer in particle-laden and dispersed multiphase environments.

[1] Physics of Fluids Group and Max Planck Center for Complex Fluid Dynamics, MESA+ Institute and J. M. Burgers Centre for Fluid Dynamics, University of Twente, P.O. Box 2177500AE Enschede, The Netherlands. [2] Center for Combustion Energy and the Department of Energy and Power Engineering, Tsinghua University, 100084 Beijing, China. [3] Max Planck Institute for Dynamics and Self-Organization, 37077 Göttingen, Germany. These authors contributed equally: Varghese Mathai, Xiaojue Zhu. Correspondence and requests for materials should be addressed to C.S. (email: chaosun@tsinghua.edu.cn) or to D.L. (email: d.lohse@utwente.nl)

The buoyancy-driven motion of particles rising or falling through a fluid has captured the interest of scientists and engineers for centuries. From the earliest observations of rising bubbles by Da Vinci to the falling hog-bladder experiments by Newton, the variability observed in the trajectories of these particles is intriguing[1,2]. Such variability can have far-reaching implications for many natural and industrial particle-laden and tethered-body flows, wherein the movement of the particles can significantly alter the drag and the transport of heat and nutrients in the fluid[3]. For example, the presence of rising bubbles and particles in the oceans can result in warmer ocean waters and larger positive temperature gradients[4–6]. Similarly, in industrial reaction catalysis, buoyant particles/bubbles are often released to enhance mixing in the fluid[7–9]. From these observations, a link can be inferred between the oscillatory motions and wakes of the buoyant particles and the processes of turbulent diffusion at work. Several studies have emphasized the importance of this[10]; however, few have established the precise factors that trigger the oscillatory dynamics.

The motion of a buoyant particle in a flow is a complex two-way coupled problem. The particle moves through the fluid in response to the flow fluctuations, and this motion in turn exerts a back-reaction on the flow. The path-oscillations that result are often robust to interactions between the neighboring particles and to changes in the flow conditions, making the studies of isolated bodies relevant to multiphase flows[11,12]. In order to model such systems, often a prototype problem is considered. The particle is modeled as a sphere, and the flow is assumed to be turbulent, homogeneous, and isotropic. For the particle dynamics, two control parameters have been proposed to be of relevance[13,14]: the particle-to-fluid density ratio $\Gamma \equiv \rho_p/\rho_f$, and the particle Galileo number $Ga \equiv \sqrt{gD^3(1-\Gamma)}/\nu$, where $g$ is the acceleration due to gravity, $D$ is the particle diameter, and $\nu$ is the kinematic viscosity of the fluid. Ga governs the development of wake-instabilities behind the particle, and $\Gamma$, the response of the particle to these wake-induced forces. Once the background turbulence is also considered, two additional parameters need to be included: the particle's size in relation to the dissipative length scale of the flow ($\Xi \equiv D/\eta$), and the Taylor Reynolds number, $Re_\lambda$, of the flow[15]. Hence, knowledge of these four dimensionless groups: [$\Gamma$, $\Xi$, $Re_\lambda$, and Ga] should seem to completely define the problem. Expecting these dependences, researchers have performed extensive studies on the dynamics of light and heavy particles in a range of flow environments[16–18].

A detail that has often been overlooked in the past is the rotational dynamics of buoyant spherical particles. Theoretical and numerical studies have mostly employed the classical Kelvin–Kirchhoff equations, expressing the conservation of linear and angular momentum for the coupled fluid-body problem[2]:

$$\left(\Gamma + \frac{1}{2} + B_U \delta\right)\frac{d\mathbf{U}}{dt} + \Gamma \mathbf{\Omega} \times \mathbf{U} = \frac{\mathbf{F}_Q}{m_f} + (\Gamma - 1)g; \quad (1)$$

$$\left(\frac{1}{10}I^* + B_\Omega \delta\right)\frac{d\mathbf{\Omega}}{dt} = \frac{\mathbf{T}_Q}{m_f D^2}; \quad (2)$$

where $\Gamma$ is the sphere mass-density ratio, $\mathbf{U}$ is the sphere velocity vector, $\mathbf{\Omega}$ is the angular velocity vector, $g$ is the acceleration due to gravity, $I^* \equiv I_p/I_f$ is the moment of inertia (MoI) ratio, where $I_p$ is the particle MoI and $I_f$ is the MoI of the fluid volume displaced by the particle, $\mathbf{F}_Q$ and $\mathbf{T}_Q$ are the fluid force and torque vectors, respectively, $m_f$ is the mass of the fluid displaced by the sphere, and $D$ is the sphere diameter. Note that $\delta = \sqrt{\frac{\nu\tau}{\pi D^2}}$ is the dimensionless Stokes boundary layer that develops in time $\tau$[19,20]. The prefactors $B_U = 18$ and $B_\Omega = 2$ are known analytically from the unsteady viscous contributions[18,21].

Equations (1) and (2) point toward the two parameter dependences, namely, the particle's mass-density ratio $\Gamma$ and its MoI ratio $I^*$. While rotation appears explicitly through Eq. (2), its effect on the particle dynamics has been ignored for isotropic bodies (spheres and cylinders). A few experiments detected the rotation for neutrally buoyant spheres in turbulence[22,23]; however, its role on the flow dynamics was not revealed. Even for freely rising spheres, the rotational effects have been largely ignored in the past[13,14]. One possible reason could be the small torque coefficients ($C_\tau \sim 10^{-4}$) reported in fixed/falling sphere studies[24]. From the torque balance for a spherical particle in a viscous fluid: $\left[I^* + k_1 \frac{1}{\sqrt{Re_p}}\right]\frac{\Delta\zeta}{\Delta t^2} = k_2 C_\tau$, one could estimate a typical rotational amplitude. Here $I^* \equiv I_p/I_f$ is the MoI ratio, where $I_p$ is the particle MoI and $I_f$ is the MoI of the fluid volume displaced by the particle. $Re_p$ is the mean particle Reynolds number, $\Delta\zeta$ is the rotational amplitude, $\Delta\tilde{t}$ is the characteristic timescale, and $k_1 = 20\sqrt{\frac{2}{\pi}}$ and $k_2 = 7.5$ are prefactors that can be determined analytically (Supplementary Discussion). The largest rotation is expected when $I^* \to 0$. In this limit, and considering $Re_p \sim \mathcal{O}(10^3)$, an order of magnitude analysis would suggest an amplitude of rotation $\Delta\tilde{\zeta} \sim 1°$ (Supplementary Discussion). Such small rotations were expected to not influence the instability onset[25]. Therefore, in most experimental studies, $\Gamma$ was varied by using various combinations of hollow and solid spheres made from a variety of materials[13,14,17]. In doing so, the particle's MoI or $I^*$ varied erratically, the implications of which have not yet been considered.

In the present work, we investigate the possibility of changing the translational dynamics of spherical particles in flows by tuning their moment of inertia (MoI). We perform experiments on three-dimensional (3D) printed spherical particles in a turbulent water flow, and track their translational and rotational motions by following a pattern painted onto the particles. We demonstrate that vigorous path-instabilities may be triggered by simply reducing a particle's MoI. A sphere with a low MoI undergoes a fluttering type of motion, while a high MoI sphere displays a tumbling type of motion. We reveal that the coupling between translation and rotation for the low MoI spheres is crucial in triggering these path-instabilities. Finally, we draw some analogies to the path-instabilities already observed for anisotropic particles such as disks, strips, and falling cards.

## Results

**Turbulence experiments.** We begin with the case of buoyant spherical particles released in a turbulent flow. Two spheres were considered, both weighing $\approx 15.2 \pm 0.1$ g, that corresponds to a mass-density ratio of $\Gamma \approx 0.88$ in water. The spheres have identical mass, diameter, and surface properties, which leave us with identical values for the particle control parameters ($\Gamma \approx 0.88$ and $Ga \approx 6000$). This mass ratio is well above the critical mass ratio ($\Gamma_{crit} = 0.6$) reported in prior studies[14,17]. At the same time, we introduce a modification to the sphere's composition. One sphere was fabricated as a hollow spherical shell, with an MoI, $I_p \approx 1.87 \times 10^{-6}$ kg m$^2$, and the other, as a thin spherical shell with a dense metallic core, with $I_p \approx 1.05 \times 10^{-6}$ kg m$^2$ (Fig. 1b). These resulted in dimensionless MoIs $I^* \equiv I_p/I_f \approx 1.0$ and 0.6, respectively. Here, $I_p$ is the particle MoI, $I_f = m_f D^2/10$ is the MoI of a fluid sphere with the same volume as the particle, and $m_f = \rho_f \pi D^3/6$ is the mass of the fluid sphere. We note that for a homogeneous sphere $I^* = \Gamma = 0.88$. Thus the values of $I^*$ we chose here ($I^* = 1.0$ and $I^* = 0.6$) are higher and lower, respectively, than the homogeneous case.

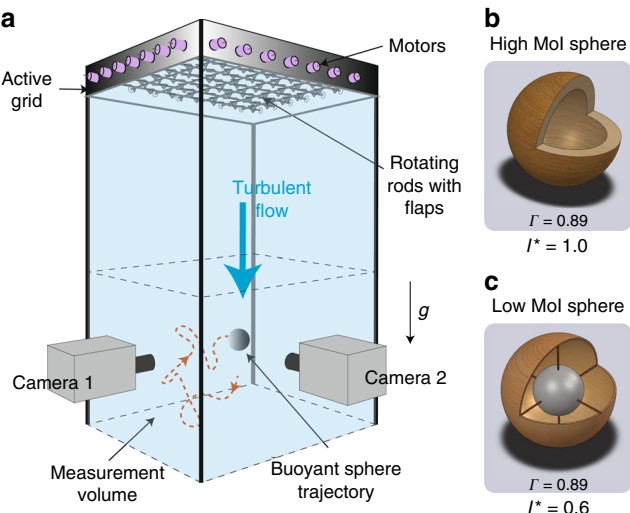

**Fig. 1** Experimental facility and spheres used in the present study. **a** A perspective representation of the measurement section of the Twente Water Tunnel (TWT) with the two camera experimental arrangement for tracking the trajectory and orientation of the buoyant spheres in turbulent flow (not to scale). **b, c** Cut-section schematics of the high and the low moment of inertia (MoI) spheres, respectively. The high MoI sphere is hollow. The low MoI sphere is constructed with a dense metallic core, which is held in the center using thin spokes as shown in **c**

The spheres were released in a turbulent flow, which was generated using an active grid located upstream to the measurement section of the Twente Water Tunnel (TWT) (Supplementary Fig. 1a). The water tunnel was configured to have a downward flow in the measurement section, with a Taylor-scale Reynolds number $Re_\lambda \approx 300$, and the particle size was a fraction of the integral scale of the turbulence ($\Xi \approx 100$). The downward mean flow speed in the measurement volume was adjusted to be comparable to the rise velocity of the particles. This ensured that the buoyant spheres were rising against the flow, enabling us to track long particle trajectories[26]. The typical duration of a trajectory was around 30 $T_L$ (or 100 $\tau_{viv}$), where $T_L$ is the integral timescale of turbulence[27], and $\tau_{viv}$ is the typical vortex-shedding timescale. In total, we recorded a duration of approximately 500 $T_L$, which gave well-converged Lagrangian statistics. The particles were imaged using two high-speed cameras placed at a 90 degree angle between them, and recorded at 500 frames per second. An analytically prescribed pattern was painted on the spheres, which enabled us to track their orientation in 3D (Supplementary Discussion and Supplementary Fig. 1b–d). This method has been validated and tested[28], and the measurement error in the detected orientation was <1°. The output of the orientation tracking method in axis-angle convention was used to obtain the instantaneous angular velocity and angular acceleration of the spheres. Thus we obtain a complete description of the particle's dynamics, composed of three-translational and three-rotational degrees of freedom.

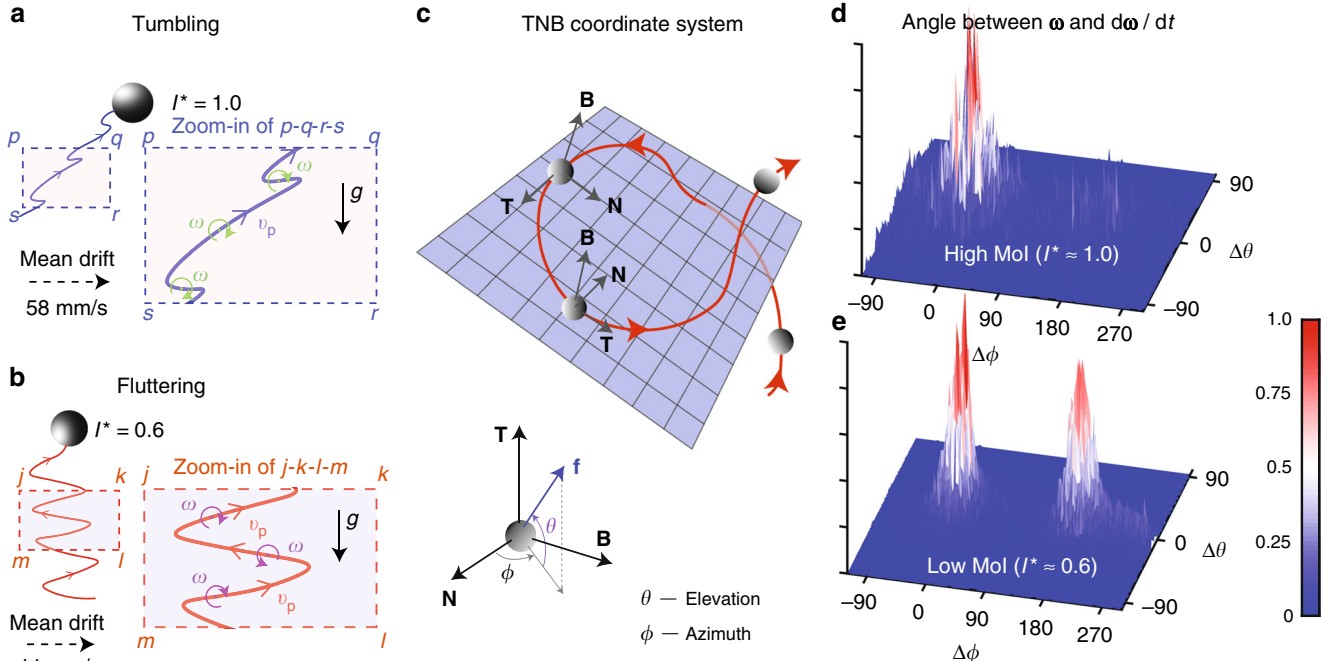

**Fig. 2** Tumbling and fluttering motions of buoyant spheres in turbulence. Drawings representative of a typical trajectory traced by the **a** high MoI sphere and the **b** low MoI sphere in a turbulent flow at $Re_\lambda \approx 300$. The zoomed-in views of the rectangular windows *pqrs* and *jklm* are also shown. The high MoI sphere tumbles in the flow, where the direction of rotation does not change much during the motion. This tumbling motion induces a mean horizontal drift for the particle. The low MoI sphere flutters in the flow, where the directions of rotation and translation undergo frequent reversals at a rate comparable to that of vortex-shedding frequency. This fluttering is thought to stabilize its motion to remain vertical to the mean. **c** Sphere motion transformed into a moving TNB coordinate system, where **T**—the direction of the particle's instantaneous velocity $\mathbf{v}_p$, **N**—direction of curvature of the particle trajectory, and **B**—the binormal vector is defined such that $\mathbf{N} = \mathbf{B} \times \mathbf{T}$. The lower figure shows the TNB coordinate system, where the orientation of an arbitrary vector **f** can be expressed in terms of the elevation $\theta$ and azimuth $\phi$ angles. **d, e** Normalized histograms of the angle between the angular velocity vector **ω** and its time derivative d**ω**/d$t$, expressed in terms of $\Delta\theta$ and $\Delta\phi$ for the **d** high and **e** low MoI spheres, respectively. Supplementary Movies 1 and 2 compare the two types of motion. The typical amplitude of flutter and tumble is 1–2 sphere diameters during an oscillation

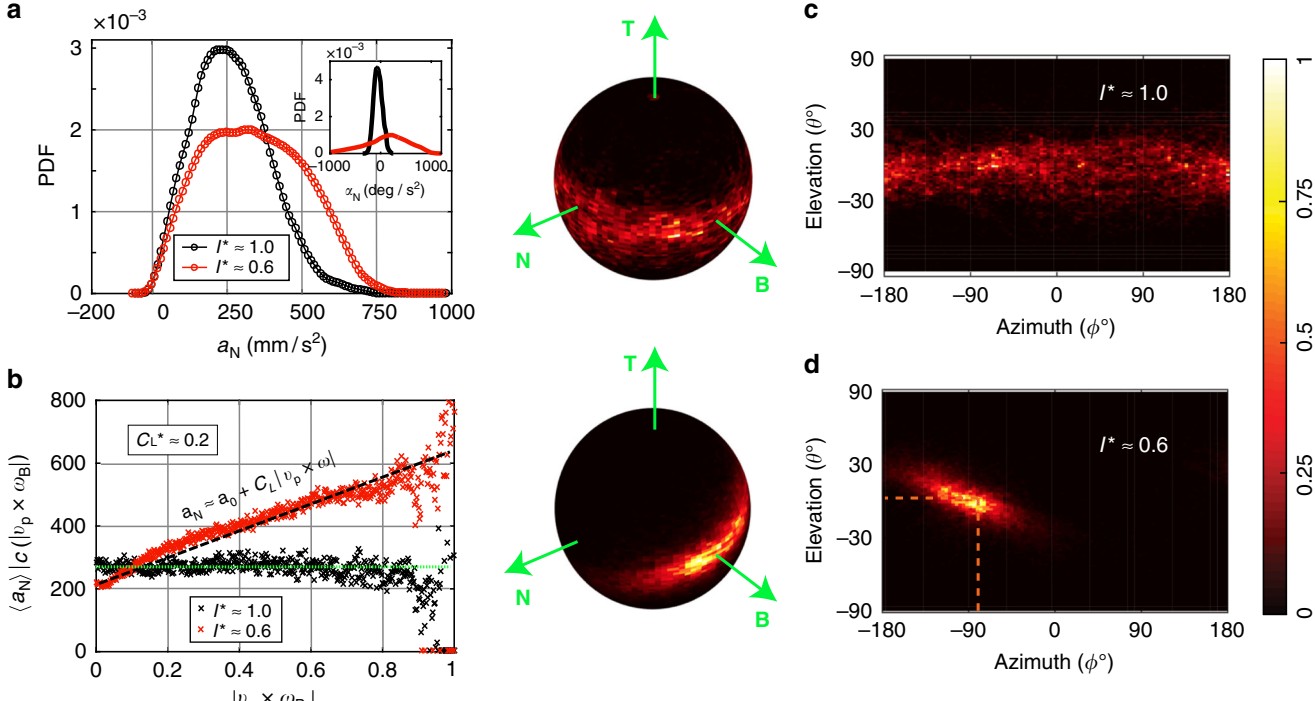

**Fig. 3** Translational acceleration statistics and rotation alignment statistics of buoyant spheres in turbulence. **a** Probability density function (PDF) of the centripetal acceleration $a_N$ for the high (in black) and low MoI (in red) spheres rising through a turbulent flow at $Re_\lambda \approx 300$. Note that $a_N$ is directed along **N** of the TNB coordinate system. Inset shows the PDF of the angular acceleration $\alpha_N$ for the two spheres. **b** $\langle a_N \rangle|_c$, the acceleration conditioned on the magnitude of lift force[32,43] $\left( \mathbf{v}_p \times \boldsymbol{\omega} \right)$ for the high (in black) and low MoI (in red) spheres. The dashed line shows the prediction for the enhancement of $\langle a_N \rangle|_c$ due to a rotation-induced lift force $C_L \left| \mathbf{v}_p \times \boldsymbol{\omega} \right|$. The dotted line shows the nearly constant value of $\langle a_N \rangle|_c$ for the high MoI sphere. Normalized histograms showing the orientation of the angular velocity vector $\boldsymbol{\omega}$ in the TNB coordinate system for the **c** high and **d** low MoI spheres in the turbulent flow. The planar histograms (right half of **c**, **d**) show the same data as the spherical histograms (left half of **c**, **d**), but in terms of the elevation $\theta$ and azimuth $\phi$ angles. The particle rotation aligns strongly with **B** for the low MoI sphere, while it is spread in the NB plane for the high MoI sphere

**Flutter-to-tumble transition**. The above settings lead to a situation where the control parameters: $\Gamma$, Ga, $\Xi$, and $Re_\lambda$, are identical for the spheres we consider. At the same time, the spheres have notably different moments of inertia ($I^* \approx 1.0$ and $I^* \approx 0.6$). Strikingly, this difference leads to dramatic changes in the translational dynamics. Evidence for this was first seen while tracking the movement of the spheres. The low MoI sphere undergoes zig-zag motions and remains in the middle of the measurement section. In contrast, the high MoI sphere shows a tendency to drift horizontally in the flow and approaches the channel walls (Fig. 2a, b). The mechanism behind these preferential movements becomes clear when we look at the particle's translational and rotational motions simultaneously (Supplementary Movies 1 and 2). The high MoI sphere is in a tumbling state, while the low MoI sphere displays a fluttering type of motion. In the tumbling state, the directions of rotation and translation do not change much during the motion (Fig. 2a). However, for fluttering, both the rotation and translation undergo periodic reversals of direction (Fig. 2b) at a rate comparable to the sphere vortex-shedding frequency (Supplementary Discussion and Supplementary Fig. 2).

Further evidence for the tumble–flutter transition can be found by viewing the particle motion from a Lagrangian (TNB) coordinate system, i.e., one that moves with the sphere (Fig. 2c). The orientation of a vector is expressed in terms of the elevation and azimuth $\theta$ and $\phi$, respectively. In Fig. 2d, e, we show the normalized histogram of the angle between the vector $\boldsymbol{\omega}$ and its time derivative $\boldsymbol{\alpha} \equiv d\boldsymbol{\omega}/dt$. Here $\boldsymbol{\omega}$ and $\boldsymbol{\alpha}$ are the angular velocity and angular acceleration vectors, respectively. An alignment between these vectors would indicate that the particle's angular

velocity increases in magnitude but without a change in the direction of rotation. An anti-alignment would mean that the angular velocity decreases but without a change in the direction of rotation. For any situation where $\boldsymbol{\omega}$ changes direction, the two vectors would not be aligned or anti-aligned. Strikingly, for the high MoI sphere, $\boldsymbol{\omega}$ is preferentially aligned with $d\boldsymbol{\omega}/dt$ (single peaked). $\boldsymbol{\omega}$ also aligns with the mean horizontal drifting direction of the particle (Supplementary Discussion and Supplementary Fig. 3), which is evidence that the sphere statistically tumbles in the direction of its horizontal drift. We believe that this tumbling motion is crucial in establishing the mean horizontal drift for the high MoI sphere. The low MoI sphere, however, shows almost equal probability for $\boldsymbol{\omega}$ and $d\boldsymbol{\omega}/dt$ to be aligned and anti-aligned (double peaked). In addition, $\boldsymbol{\omega}$ does not preferentially align with the mean horizontal drifting direction of the particle (Supplementary Discussion and Supplementary Fig. 3). Thus a fluttering type of motion occurs, which stabilizes the low MoI sphere to remain in the bulk of the water channel flow. An analogy may be drawn to falling disks, strips, and paper, where a similar tumble–flutter transition has already been observed owing to MoI reduction[29,30] (Supplementary Movie 3). However, some form of geometrical anisotropy (of the particle) was considered necessary to induce this transition[31]. The present finding demonstrates that such transitions are possible even for an isotropic body such as a sphere.

**Acceleration statistics**. The tumble–flutter transition revealed that the high MoI sphere drifts predominantly along a particular direction, whereas the low MoI sphere (fluttering) undergoes frequent reversals in its direction of motion (Supplementary

Fig. 3 and Supplementary Movies 1 and 2). The fluid forces responsible for the differences in the dynamics may be gauged from the particle's acceleration statistics. Figure 3a shows the probability density function (PDF) of the centripetal acceleration $a_N$ for the two spheres under consideration. $a_N$ is always directed along **N** of the TNB coordinate system. A reduction of MoI produces a significant change in the shape of the PDF, with a mean value that is almost double to that of the high MoI sphere. When we compare the angular accelerations, the difference is even more dramatic (inset to Fig. 3a). If the torques acting on the spheres were comparable, one would expect the angular acceleration to be increased by ~66% ($\mathcal{T} = I_p d^2\theta/dt^2$). Instead, the root mean square of angular acceleration is increased by ~430%. Such a dramatic increase in the angular acceleration can only be caused by an enhancement in the torque. Clearly, rotation plays a role in changing the torques acting on the particle, which in turn leads to enhanced translational accelerations. This is quantified in Fig. 3b, where we show that the acceleration $a_N$ is conditioned on the magnitude of a rotation-induced lift (or Magnus) force[32] $F_L \propto \mathbf{v}_p \times \boldsymbol{\omega}$. For the high MoI sphere, $a_N$ appears to be almost independent of the Magnus force. However, a clear dependence is seen for the low MoI sphere. The origin of this becomes clear once we plot the orientation of $\boldsymbol{\omega}$ in the TNB coordinates, (Fig. 3c, d). For the low MoI sphere (Fig. 3d), $\boldsymbol{\omega}$ aligns along the **B** direction, resulting in an alignment of $-\mathbf{v}_p \times \boldsymbol{\omega}$ with the direction of $a_N$. Therefore, the acceleration $a_N$ may be written as $a_N \approx a_0 + C_L|v_p \times \omega|$, as shown by the dashed line in Fig. 3b. A lift coefficient $C_L \approx -0.2$ provides a fair prediction of the increase in $a_N$. A negative $C_L$ of this value is expected[32], since we have a buoyant particle that rises and rotates in the flow[33,34]. For the high MoI sphere (Fig. 3c), $\boldsymbol{\omega}$ lies in the NB plane with no preferential orientation, which explains the absence of correlation in the conditioned acceleration plot, Fig. 3b. Thus tuning the MoI has enabled us to alter both the fluid forces and torques acting on a spherical particle, leading to an overall modification of its dynamics in turbulence. Whether these effects are restricted to a turbulent flow or not is unclear. To address this, we will next explore the dynamics of buoyant spheres rising in an undisturbed flow setting, i.e., in quiescent liquid.

**Free-rise experiments**. Figure 4 shows the free-rise trajectories and wake patterns of the high and low MoI spheres rising through still fluid (Ga ≈ 6000 and Ga ≈ 500). For this experiment, the spheres were released in a glass tank of $280 \times 280$ mm$^2$ cross-section and 1500 mm height. We inject a patch of sodium fluorescein dye, which is the sodium salt of fluorescein ($C_{20}H_{10}Na_2O_5$). The dye was injected just above the sphere, near its release position at the base of the water tank (Supplementary Discussion and Supplementary Fig. 4). Once the sphere is released, it rises through the dye, entraining a part of the dye with it, and also shedding some dye in its wake as it rises. The dye fluoresces for blue illumination (≈490 nm wavelength), which helps visualize the wake. The intensity of the dye represents the relative concentration. Note that this does not represent the absolute vorticity in the wake but only gives a qualitative picture of the wake pattern, similar to the wake patterns reported in prior studies[14,17]. At the high Ga (Ga ≈ 6000 in Fig. 4a), the wake is turbulent. The high MoI sphere (left) rises with a small amplitude of oscillation and a nearly vertical wake pattern. When $I^*$ is reduced, large amplitude oscillations are triggered, and we witness a spread-out wake pattern behind the particle (right). Similarly, at a lower Ga (Fig. 4b), the oscillation amplitude is enhanced, and the wakes differ in the spread and the number of structures shed per oscillation. A similar effect was recently observed by us in numerical simulations of two-dimensional circular cylinders rising in still fluid[35] (Supplementary Figs. 5 and 6 and Supplementary Movie 6). Presumably, the wake modification due to particle rotation is crucial to these changes in the oscillation amplitude. Interestingly, in the free-rise experiments, we do not observe the tumbling motions. It is likely that the flow perturbations from the incident turbulence are necessary to trigger these tumbling events.

## Discussion

Comparing the sphere and (two-dimensional) cylinder dynamics has shed light on some key aspects of the buoyancy-driven motions of isotropic bodies in general. A long-standing debate in this subject relates to the existence a critical mass-density ratio ($\Gamma = 0.6$ for sphere[14], and $\Gamma = 0.54$ for cylinder[36]), marking the onset of large amplitude vibrations for freely rising spheres and cylinders). While this has been reported in literature, the physical mechanism behind such a sharp transition has remained puzzling. The main question remains as to how a marginal reduction in mass density ($\Gamma$) could give rise to a significant enhancement in vibration amplitude, since the effective mass of the system (i.e., actual+added mass) changes by only a small factor[37]. Our observations suggest that one possible reason for this could lie in the differences in the particle's rotational dynamics, which manifest through its rotational inertia (since $I^*$ changes along with $\Gamma$ in most situations). This new perspective might also explain the origin of the wide variation that was witnessed in prior experimental studies on freely rising spheres[13,14] and cylinders[36].

On a different note, the insights gained here may also be of relevance to the rising motions of spherical bubbles in water. It is well known that, when the bubble surface is contaminated, it spirals or zig-zags[38], while the same bubble in pure water rises vertically straight[39] (up to a certain larger Galileo number). The clean bubble, owing to its free-slip boundary condition, obviously does not rotate. In contrast, the contaminated bubble can rotate due to a combination of no-slip at the interface and low rotational inertia ($I^* \to 0$). These could be the contributing factors to the observed differences in the onset behavior of their path-instabilities[40,41]. For non-spherical particles (and deformable bubbles), the role of MoI might be even more crucial. For a non-

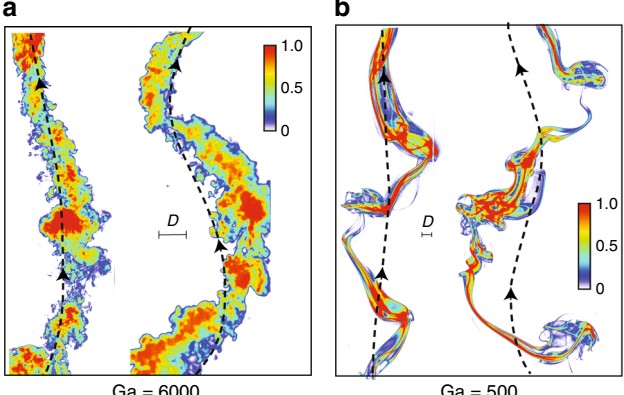

**Fig. 4** Free-rise experiments of buoyant spheres. **a** Trajectories and wakes of high (left) and low (right) MoI spheres at Ga ≈ 6000. **b** Trajectories and wakes of high (left) and low (right) MoI spheres at Ga ≈ 500. In **a**, **b**, the wakes were visualized by releasing a fluorescent dye. False coloring was adopted based on the intensity of the dye in the wake. Note that the diameter scale 'D' is shown in each case. Reducing the MoI or allowing rotation at fixed $\Gamma$ triggers an increase in the oscillation amplitude along with a change in the wake pattern. Supplementary Movies 4 and 5 correspond to the cases shown in **a**, **b**, respectively

spherical particle, the torque on the particle arises from two contributions: from skin friction forces and pressure forces[2]. Additionally, the particle's angle of attack changes as it rotates, resulting in significant torques. Thus strong rotational motions can be expected, and hence the MoIs might have a major role in observed particle dynamics. The coupled translational and rotational motions of non-spherical particles (oblate and prolate ellipsoids) will be the focus of a future investigation.

In conclusion, the use of simultaneous 3D particle position and orientation tracking has enabled us to resolve the coupled translational and rotational dynamics of buoyant spherical particles in a range of flow environments. We have shown that the onset of path-instability can be closely linked to the rotation of the particle and that resonance may be induced (or inhibited) by tuning the particle's MoI. This radically changes the way one perceives the dynamics of buoyancy-driven isotropic bodies, and opens the door for control of path- and wake-instabilities by tuning the rotational inertia. The concept could be exploited in chemical engineering processes, where the mixing and transport of nutrients can be effectively enhanced through the introduction of vigorously vibrating spheres[7]. Further, the flutter–tumble transition we observed for buoyant spheres in turbulence offers a few flow-control opportunities. For instance, particles that migrate toward channel walls could be used to modify the near-wall flow structure[3]. By tuning the MoI, one could design spheres that accumulate near the walls, with potential for drag/heat transfer modifications in dispersed multiphase flow environments. Other avenues of application could lie in sports ballistics, where zig-zag and spiral trajectories add to the unpredictability of the game. While this has historically been achieved by introducing surface heterogeneities and/or spin to the ball[42], rotational inertia could be used as an additional lever to trigger such path-instabilities, thereby lending richer diversity to various ball sports.

## Methods

**Experimental methods**. The experiments were performed in the TWT facility, in which an active grid generates nearly homogeneous and isotropic turbulence in the measurement section. The water tunnel was configured to have a downward flow in the measurement section, and the Taylor Reynolds number of the flow $Re_\lambda \approx 300$. We used a high-precision 3D printer (Rapidshape S30L) to fabricate the hollow spherical particles. The surface roughness was within 25 µm, and the spheres were symmetric for any plane passing through their geometric center. Rolling and floating tests were used to check for any inconsistencies. The particles were imaged using two high-speed cameras placed at a 90 degree angle between them (Supplementary Discussion and Supplementary Fig. 1). An analytically prescribed pattern was painted on the spheres.

**Image processing**. The particles were detected using the Circular Hough Transform method, which was implemented using the imfindcircles function in MATLAB. The projection of the painted pattern was compared with the synthetic image to retrieve the orientation (Supplementary Fig. 1c,d). Combining the two detection methods, we can track six degrees of freedom for the sphere released in the turbulent flow.

**Data availability**. The data that support the findings of this study are available from the authors upon reasonable request.

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

## Acknowledgements

V.M. is grateful to L. Olde Olthof for significant help during the experiments. We thank P. van der Plas, S. Maheshwari, J. Will, S. Wildeman, B. Verhaagen, and M. van Limbeek for useful discussions. We thank G.-W. Bruggert, D. van Gils, and M. Bos for the technical support. This work was supported by the Natural Science Foundation of China under the Grant No. 11672156, the STW foundation of the Netherlands, COST action MP1305, and the European High-Performance Insfrastructures in Turbulence (EuHIT).

## Author contributions

V.M., C.S., and D.L. designed the research. V.M. performed the experiments. V.M. and X. Z. performed the analysis. All authors discussed the results and contributed to the writing of the manuscript.

## Additional information

**Competing interests:** The authors declare no competing interests.

