## [Peer Review File · Nature Communications]

Reviewer #1 (Remarks to the Author):

This paper describes the changes in the trajectory and wake of sedimenting spheres in viscous Newtonian flows. This subject has been vastly studied in the fluid mechanics community. It is a paradigmatic flow to study fluid-structure interaction, turbulence, multiphase flows, etc. As it well described by the authors, even the simplest case of a sphere sedimenting freely is not well understood. In my opinion, the study of this subject has been long stagnant because the lack of progress and the apparently contradictory results. The present paper proposes, finally, a new direction! Indeed, no one before had taken into consideration the effect of the moment of inertia of particles in the dynamics of the motion of spheres. Because of this fundamental idea, I do believe that the present investigation may be worthy of being published Nature Communications.

I have identified a number of issues that I would like the authors to address before I can recommend publication:

- To begin with, the title may be misleading. The authors, in my opinion, do not control the path instability. What they do is to change the point at which it occurs, by varying the moment of inertia. I think that the title could be changed to avoid confusion.
- It would be of great value to try to extend the arguments presented in this paper to try to understand the path instabilities for the case of bubbles. In such a case, the moment of inertia would be zero. Would the arguments expressed in lines 76-82 still hold? How would the non-spherical particle shape (for either bubbles or non-spherical particles) would affect the moment of inertia argument? Adding a discussion in this regard would greatly increase the possible impact of the paper.

Other minor comments:

- It is not obvious how I_f (the moment of inertia of the fluid) is calculated. Please add the calculation for the case of the sphere.
- Please elaborate on the Lagrangian TNB coordinate system. Figure 2-C is confusing: from the text \vec{V} and T are supposed aligned, but the inferior image seems to indicate otherwise.
- Line 128. α is a vector, so to be consistent with the notation, it should have an arrow on top.
- It is not clear how Figures 3(c) and (d) were obtained. Please elaborate. Do the spheres and planes show the same information?
- Please define the centripetal acceleration a_N . How is it calculated?
- It is not clear how the wakes are visualized and what is the information conveyed by the color maps in figs 4 and 5. Please clarify. What do the colors mean?

Reviewer #2 (Remarks to the Author):

This is a highly innovative study, both scientifically and technically. The asymmetric behaviors of heavy particles falling straight in liquid and light bubbles or buoyant particles rising in a fluid in a zigzag or spiraling way has long been a puzzle. Mathai et al. have come to recognize that the parameter space used in all the previous studies was in fact incomplete. Using both experimental and numerical approaches, these authors have shown that the rotational inertia or moment of inertia (which controls the rotational behavior of particles) plays a key role in the falling or rising of spherical objects. By designing and fabricating particles with different moment of inertia but with the same shape, size and mass, the authors have found something amazing, that the translational dynamics for particles with different moment of inertia (MI) are very different, the ones with high MI drift in the flow, whereas the ones with low MI zigzag in the flow with their rotation axis reverses frequently. By doing numerical simulation, the authors found that zigzag behavior and

the associated rotation axis reversal may be related to the oscillating vortex street that is generated for turbulent flows past a bluff body, which usually occurs with the bluff body fixed in the flow. This brings the current understanding of fluid-solid interaction into a new level. This study not only solves a long mystery but also has potential applications in many industrial processes involving particle laden or multiphase flows. As suggested by the authors, this opens a way of controlling the dynamics of spherical particles by tuning their moment of inertia. I think the paper deserves publication in Nature Communications. But before I can recommend its publication, I would like the authors to address the questions below:

- 1) What is the logic behind the current choice of values for mass density ratio Γ , and for moment of inertia ratios I^* . Namely, $\Gamma = 0.89$, and $i^* = 1.0$ and 0.6 .
- 2) You need long particle trajectories to obtain many of the statistics. How can you obtain long trajectories from the turbulence experiment without the spheres moving out of view?
- 3) How is the free-rise experiment conducted? I don't see details of how the release is conducted. Is the dye injected from inside the sphere?
- 4) For the free rise experiment, how can you be sure the oscillations are in a plane parallel to the field of view.
- 5) Why don't we see tumbling in the free rise experiment?
- 6) Does the lift coefficient reported in Fig 3 (B) make sense for the Reynolds number of the experiment? Why is C_L negative?
- 7) In Fig. 2(a) inset, with decrease in I^* , the angular acceleration increases much more than suggested by simply equating the torques. What might be the physical origin of this?
- 8) What is the amplitude of flutter and tumble relative to the particle diameter? Can the authors quantify these?
- 9) What is the error in the sphere rotation detection, in terms of standard deviation of error in degrees?
- 10) Why there are no clear vortices visible in left figure of the free rise experiments? Can the dye visualization be used to quantify the vorticity?
- 11) The authors state that for low MI particles, the reversal frequency of the rotation axis and related fluttering motion is similar to the vortex shedding frequency found for flow past spheres, as shown by their numerical simulation. Yet, I see that the simulation is for cylinders not spheres as in the experiment. Why is this?
- 12) A related question is: can the authors perform 3D simulations of the tumbling/fluttering motions?

NCOMMS-17-32094-T by Mathai, Zhu, Sun, Lohse

February 12, 2018

1 Response to Referee 1

General comments:

Comment: *This paper describes the changes in the trajectory and wake of sedimenting spheres in viscous Newtonian flows. This subject has been vastly studied in the fluid mechanics community. It is a paradigmatic flow to study fluid-structure interaction, turbulence, multiphase flows, etc. As it well described by the authors, even the simplest case of a sphere sedimenting freely is not well understood. In my opinion, the study of this subject has been long stagnant because the lack of progress and the apparently contradictory results. The present paper proposes, finally, a new direction! Indeed, no one before had taken into consideration the effect of the moment of inertia of particles in the dynamics of the motion of spheres. Because of this fundamental idea, I do believe that the present investigation may be worthy of being published Nature Communications.*

Response: We thank the reviewer for appreciating the novelty of our work. We are glad the reviewer recommends the paper to be published in Nature Communications. Below we have addressed the remaining questions and concerns of the reviewer.

Comment: *To begin with, the title may be misleading. The authors, in my opinion, do not control the path instability. What they do is to change the point at which is occurs, by varying the moment of inertia. I think that the title could be changed to avoid confusion.*

Response: We agree with the referee. We now changed the title to:

"Flutter to tumble transition of buoyant spheres triggered by rotational inertia changes".

We hope this gives a better representation of the storyline of the paper.

Comment: *It would be of great value to try to extend the arguments presented in this paper to try to understand the path instabilities for the case of bubbles. In such a case, the moment of inertia would be zero. Would the arguments expressed in lines 76-82 still hold? How would the non-spherical particle shape (for either bubbles or non-spherical particles) would affect the moment of inertia argument? Adding a discussion in this regard would greatly increase the possible impact of the paper.*

Response: We are glad the reviewer likes the theoretical arguments we presented in lines 76-82. One of the motivations for presenting this analysis along with the limiting case of $I^* \rightarrow 0$ was to point to the situation of a rising bubble. Before going into the case of a deformable bubble, we consider the simpler example of a rising millimetric bubble (nearly spherical) in water. It is well-known that when the bubble surface is contaminated it spirals or zig-zags, while the same bubble in pure water rises vertically straight (up to some larger Galileo number). It could very well be that the different rotational motions – the clean bubble does not rotate because of "free-slip" boundary condition, while the contaminated one can rotate due to its "no-slip" condition and low rotational inertia – might contribute to the differences in the onset behavior of the path-instability.

Now let us consider the case of a rising deformable bubble, or a non-spherical particle such as an oblate spheroid. In these cases the rotational inertia will have an even greater influence, and can lead to vigorous path-oscillations. For a non-spherical particle (or a deformed bubble), the torque on the particle/bubble arises from two contributions: (1) from skin friction forces, and (2) from

pressure forces (note that the contribution from pressure forces is zero for a spherical particle). Additionally, due to the non-spherical shape, the particle's boundary domain changes as it rotates, resulting in a change in the particle's "angle of attack". At certain angles of attack, the torque on the particle can be significant. Thus, strong rotational motions can be expected, and hence the moment of inertia will have a major role in the particle dynamics. In fact, the focus of a new project is on such anisotropic particles, e.g. 3D printed oblate spheroids whose boundary domain resemble the shape and aspect ratio of 2–3 mm air bubbles (although not deformable). Preliminary observations suggest some similarities to the trajectory and rotation of a rising deformable bubble.

A discussion of the above points has been included in the revised manuscript.

Minor comments:

Comment: *It is not obvious how I_f (the moment of inertia of the fluid) is calculated. Please add the calculation for the case of the sphere.*

Response: I_f is the moment of inertia of a fluid sphere that has the same diameter as the spherical particle. $I_f = m_f D^2 / 10$, where m_f is the mass of the fluid contained in the sphere's volume, i.e. $m_f = \rho_f \pi D^3 / 6$, where ρ_f is fluid density and D the sphere diameter. We have included this in the revised manuscript.

Comment: *Please elaborate on the Lagrangian TNB coordinate system. Figure 2(C) is confusing: from the text \vec{V} and \vec{T} are supposed aligned, but the interior image seems to indicate otherwise.*

Response: We apologize for the misleading figure. The reviewer is correct! The particle velocity is along \vec{T} vector. In fact, \vec{V} refers to an arbitrary "vector" that is under consideration. It was a coincidence that \vec{V} could also stand for particle velocity.

In the revised manuscript, we have renamed the vector as \vec{f} , where \vec{f} can be any arbitrary vector under consideration. \vec{f} could be particle velocity \vec{v} , or particle acceleration \vec{a} , or angular velocity $\vec{\omega}$.

Comment: *Line 128. α is a vector, so to be consistent with the notation, it should have an arrow on top.*

Response: The vector symbol has been added in the revised manuscript.

Comment: *It is not clear how Figures 3(c) and (d) were obtained. Please elaborate. Do the spheres and planes show the same information?*

Response: Figure 3(c) shows the orientation of the angular velocity vector $\vec{\omega}$ in the TNB coordinate system. Yes, the "planes" figure and the "sphere" figure are essentially the same. In the "planes" figures, the orientation of $\vec{\omega}$ is expressed in terms of elevation and azimuth angles that were defined earlier in Fig. 2(c). The sphere histogram figures were included to give a physical feel of the direction of the $\vec{\omega}$ in the TNB coordinate system (which might be difficult to interpret from just the "planes" figures).

An explanation has been included in the revised paper.

Comment: *Please define the centripetal acceleration a_N . How is it calculated?*

Response: According to the definition of TNB coordinate system, the total acceleration of the particle can be decomposed into two components: (1) due to a change of the magnitude of the particle velocity vector v_p , and (2) due to a change in the direction of v_p . The former gives the tangential acceleration a_T along the \vec{T} direction, and the latter gives the centripetal acceleration a_N , directed along the instantaneous curvature of the trajectory (\vec{N} direction). We have clarified these in the revised manuscript.

Comment: *It is not clear how the wakes are visualized and what is the information conveyed by the color maps in figs 4 and 5. Please clarify. What do the colors mean?*

Response: The wakes are visualized using sodium fluorescein dye [1, 2], which is the sodium salt of fluorescein ($C_{20}H_{10}Na_2O_5$). The setup is illuminated with blue lights (490 nm wavelength). The dye was injected just above the sphere near its release position at the base of the water tank (see supplemental material). Once the sphere is released, it rises through the dye, entraining a part of the dye in the wake, and also shedding some dye in its wake as it rises. The color of the dye you see in the images is only a representation of the relative concentration of dye at that point. This approach was chosen in the past by other researchers (e.g Horowitz and Williamson [3]). We wish to clarify that this is only a qualitative representation of the wake pattern, and not the absolute vorticity. We now mentioned these in the revised manuscript. At the lower Ga though (Fig. 4(B)), the dye can clearly visualize the vortex ring structure in the wake. We have included a discussion in the revised manuscript.

We thank the reviewer for his/her suggestions which improved our manuscript. We hope the paper will now be given full recommendation for publication in Nature Communications.

References

- [1] M. Horowitz and C. H. K. Williamson. Vortex-induced vibration of a rising and falling cylinder. *J. Fluid Mech.*, 662:352–383, 2010.
- [2] E. Alm eras, F. Risso, V. Roig, S. Cazin, C. Plais, and F. Augier. Mixing by bubble-induced turbulence. *J. Fluid Mech.*, 776:458–474, 2015.
- [3] M. Horowitz and C. H. K. Williamson. Critical mass and a new periodic four-ring vortex wake mode for freely rising and falling spheres. *Phys. Fluids*, 20(10):101701, 2008.

February 12, 2018

1 Response to Referee 2

Comment: *This is a highly innovative study, both scientifically and technically. The asymmetric behaviors of heavy particles falling straight in liquid and light bubbles or buoyant particles rising in a fluid in a zigzag or spiraling way has long been a puzzle. Mathai et al. have come to recognize that the parameter space used in all the previous studies was in fact incomplete. By designing and fabricating particles with different moment of inertia but with the same shape, size and mass, the authors have found something amazing, that the translational dynamics for particles with different moment of inertia (MI) are very different, the ones with high MI drift in the flow, whereas the ones with low MI zigzag in the flow with their rotation axis reversing frequently. This brings the current understanding of fluid-solid interaction into a new level. This study not only solves a long mystery but also has potential applications in many industrial processes involving particle laden or multiphase flows. As suggested by the authors, this opens a way of controlling the dynamics of spherical particles by tuning their moment of inertia. I think the paper deserves publication in Nature Communications.*

Response: We thank the reviewer for the nice words about our work, and for appreciating the novelty and innovation of the study. We are glad the reviewer recommends the paper to be published in Nature Communications. Below we have addressed the remaining questions of the reviewer.

Comment: *What is the logic behind the current choice of values for mass density ratio Γ , and for moment of inertia ratios I^* . Namely, $\Gamma = 0.89$, and $I^* = 1.0$ and 0.6 .*

Response: According to one of the most comprehensive experimental studies in literature (Horowitz and Williamson [1, 2]), the particle's mass ratio Γ alone determines the wake pattern and path-instability (for a fixed Galileo number). They found that $\Gamma_{crit} = 0.6$ is the critical mass ratio, below which path oscillations are seen, and above which buoyant spheres rise straight. However, in arriving at this conclusion, they used both hollow and solid spheres in their experiments, which presumably led to variations in the particle's moment of inertia ratio I^* . This realization motivated us to study a spherical particle with Γ well above the critical value of 0.6. We chose $\Gamma = 0.89$. At the same time, by designing two spheres, one hollow and the other with a dense metallic core, we obtained $I^* = 1.0$ and $I^* = 0.6$, respectively. Now, according to Horowitz and Williamson [1], both spheres should rise vertically without path-oscillations since $\Gamma = 0.89 (> \Gamma_{crit})$. This is indeed the case for the high moment of inertia (MoI) sphere ($I^* = 1.0$), which does not show path-oscillations. In contrast, the low MoI particle showed vigorous zig-zag motions.

In essence, by choosing the above combinations of Γ and I^* , we have obtained clear evidence for vigorous path-oscillations for a particle with a mass ratio $\Gamma = 0.89$ that is well above the critical Γ proposed in literature [1]. At the same time, the moment of inertia I^* needs to be low for the path-instability to occur. Therefore, the main control parameter governing path-oscillations is not the mass ratio of spherical particle, but instead, its rotational inertia I^* . A discussion on the logic behind the choice of Γ and I^* has been included in the revised manuscript.

Comment: *You need long particle trajectories to obtain many of the statistics. How can you obtain long trajectories from the turbulence experiment without the spheres moving out of view?*

Response: The present experiments in turbulence are possible due to the many unique features of the Twente Water Tunnel (TWT) facility. Firstly, the TWT has a vertically long measurement section with an active grid to control the level of turbulence. Furthermore, the flow controllability of the water tunnel allowed us to adjust the downward flow speed to match the rise velocity of the buoyant particles. These enabled us to obtain long particle trajectories since the particles do not leave the laboratory-fixed field of view. The typical duration of a trajectory was $30 T_L$ (or $100 \tau_{viv}$), where T_L is the integral timescale of the turbulence, and τ_{viv} the typical vortex shedding timescale. In total, we recorded for approximately $500 T_L$, which gave us unprecedented Lagrangian statistics. The above points have been included in the revised manuscript.

Comment: *How is the free-rise experiment conducted? I don't see details of how the release is conducted. Is the dye injected from inside the sphere? For the free rise experiment, how can you be sure the oscillations are in a plane parallel to the field of view.*

Response: The free rise experiment is conducted by releasing the sphere inside a water tank of $280 \text{ mm} \times 280 \text{ mm}$ cross section and 1500 mm high (see supplemental material). The tank was filled with either water or a mixture of glycerine and water. The viscosity of the mixture was monitored at regular intervals using a rheometer (Anton Paar MCR 102). The sphere was held to the bottom of the tank by applying suction. We injected sodium fluorescein dye [3, 4] just above the sphere in its release position. The dye, which basically is the sodium salt of fluorescein ($\text{C}_{20}\text{H}_{10}\text{Na}_2\text{O}_5$), fluoresces to blue illumination (490 nm wavelength). The dye was prepared with a concentration $\approx 5 \times 10^{-3} \text{ mol/L}$. The dye has a molecular diffusion coefficient in water, $D_m = 4 \times 10^{-10} \text{ m}^2/\text{s}$ at 22° , which is much smaller than the water kinematic viscosity ν . This leads to a high Schmidt number, $\text{Sc} \equiv \nu/D_m \approx 2500$. A camera was positioned orthogonally facing a side-wall of the glass tank. The sphere rose through the liquid and entrained the dye on its way up. As the sphere rises, the entrained dye gets gradually shed, enabling us to visualize the wake. This gave a qualitative picture of the wake structure. We have ensured that the images presented here correspond to oscillations predominantly in a plane parallel to the field of view.

A detailed description along with a schematic is included in the revised supplemental material.

Comment: *Why don't we see tumbling in the free rise experiment?*

Response: This could be due to two reasons: (a) the field of view in Fig 4 is not large enough to observe tumbling. (b) The turbulence has some role in inducing background perturbations which triggers the tumbling. We believe (b) is the probable reason, since the turbulent eddies can induce initial rotations which persist during the tumbling. We included a discussion on this in the revised manuscript.

Comment: *Does the lift coefficient reported in Fig 3 (B) make sense for the Reynolds number of the experiment? Why is C_L negative?*

Response: Yes. The lift coefficient is comparable to some fixed sphere studies from literature [5, 6]. Indeed, the lift coefficient reported in most prior studies is positive. However, we note that most of these addressed uniform flow past a rotating sphere with its center fixed. In this frame of reference, C_L is positive [7]. In the present experiment, we have a sphere moving and rotating in a flow [5, 8], in which case C_L ought to be negative, similar to the case of Magnus effect for a sport ball. A discussion on these is included in the revised manuscript.

Comment: *In Fig. 2(a) inset, with decrease in I^* , the angular acceleration increases much more than suggested by simply equating the torques. What might be the physical origin of this?*

Response: This is one of the most interesting and surprising results of the paper. Normally, when rotational inertia is reduced, one would expect the angular acceleration α to increase as $\alpha \propto 1/I^*$. This means that we can expect a 66% increase in α . However, we see up to 430% increase in α . As we explain in the paper, such a dramatic increase in angular acceleration can only be caused by an enhancement of the torque coefficient C_τ . The coupled translational and rotational motion changes the flow around the particle dramatically, leading to this increased C_τ . We have included a discussion in the revised manuscript.

Comment: *What is the amplitude of flutter and tumble relative to the particle diameter? Can the authors quantify these?*

Response: The amplitude of flutter and tumble are around 1– 2 sphere diameters. This is typical of vortex induced vibrations. We have mentioned this in the revised manuscript.

Comment: *What is the error in the sphere rotation detection, in terms of standard deviation of error in degrees?*

Response: The standard deviation of error in orientation detection is less than 1° . This is detailed in our method paper (Mathai *et al.*, Exp. Fluids 57, 51 (2016) [9]). We have given the details in the revised paper.

Comment: *Why there are no clear vortices visible in left figure of the free rise experiments? Can the dye visualization be used to quantify the vorticity?*

Response: The wakes are visualized using fluorescein sodium dye with blue light illumination (490 nm wavelength). The dye was injected just above the sphere at its release position. Once the sphere is released, it rises through the dye, entraining part of the dye in its wake, and also shedding some as it rises. The intensity of dye you see in the images is only a representation of the concentration of dye at that location. We wish to clarify that this is only a qualitative representation of the wake pattern, and not the absolute vorticity. At high Ga, the vortices cannot be very clear, as was also stated in prior studies which used the same technique [1]. At a lower Ga though, the dye pattern does show clear signatures of the wake vortices. A discussion is included in the revised manuscript.

Comment: *The authors state that for low MI particles, the reversal frequency of the rotation axis and related fluttering motion is similar to the vortex shedding frequency found for flow past spheres, as shown by their numerical simulation. Yet, I see that the simulation is for cylinders not spheres as in the experiment. Why is this? A related question is: can the authors perform 3D simulations of the tumbling/fluttering motions?*

Response: Fully resolved 3D simulations of buoyant spheres at this Reynolds number ($Re \approx 10^4$) are extremely difficult to perform using the resources we have. To our knowledge, one of the highest Re numerical simulations of light rising spheres is for $Re \approx 1000$ by Auguste and Magnaudet [10]. The tumbling fluttering simulations are prohibitively expensive in numerical simulations.

We thank the reviewer for his/her suggestions which improved our manuscript. We hope the paper will now be given full recommendation for publication in Nature Communications.

References

- [1] M. Horowitz and C. H. K. Williamson. Critical mass and a new periodic four-ring vortex wake mode for freely rising and falling spheres. *Phys. Fluids*, 20(10):101701, 2008.
- [2] M. Horowitz and C. H. K. Williamson. The effect of reynolds number on the dynamics and wakes of freely rising and falling spheres. *J. Fluid Mech.*, 651:251–294, 2010.
- [3] M. Horowitz and C. H. K. Williamson. Vortex-induced vibration of a rising and falling cylinder. *J. Fluid Mech.*, 662:352–383, 2010.
- [4] E. Alm eras, F. Risso, V. Roig, S. Cazin, C. Plais, and F. Augier. Mixing by bubble-induced turbulence. *J. Fluid Mech.*, 776:458–474, 2015.
- [5] E. Loth. Lift of a spherical particle subject to vorticity and/or spin. *AIAA J.*, 46(4):801–809, 2008.

- [6] E. Loth and A. J. Dorgan. An equation of motion for particles of finite reynolds number and size. *Environ. Fluid Mech.*, 9(2):187–206, 2009.
- [7] J. J. Bluemink, D. Lohse, A. Prosperetti, and L. Van Wijngaarden. Drag and lift forces on particles in a rotating flow. *J. Fluid Mech.*, 643:1–31, 2010.
- [8] C. Clanet. Sports ballistics. *Annu. Rev. Fluid Mech.*, 47:455–478, 2015.
- [9] V. Mathai, M. W. M. Neut, E. P. van der Poel, and C. Sun. Translational and rotational dynamics of a large buoyant sphere in turbulence. *Exp. Fluids*, 57(4):1–10, 2016.
- [10] F. Auguste and J. Magnaudet. Path oscillations and enhanced drag of light rising spheres. (*submitted manuscript*).

Reviewer #1 (Remarks to the Author):

I am satisfied by the authors replies and amendments to the paper. I think this will be an important paper. Only one small suggestion. In the section where the implications of the present results to the motion of bubbles, there are no references. The authors make very specific claims but provide no citations. Adding one or two references to classic papers in this subject would be important.

Reviewer #2 (Remarks to the Author):

The authors have satisfactorily addressed all my comments and questions. The revised version makes many points much clearer and is more readable. I think the new title is more informative. As already stated in my last report, this is an important contribution to the fields of fluid mechanics and multiphase flows. I now fully support the paper's publication in Nature Communications.

NCOMMS-17-32094A by Mathai, Zhu, Sun, Lohse

March 20, 2018

1 Response to Referee 1

General comments:

Comment: *I am satisfied by the authors replies and amendments to the paper. I think this will be an important paper. Only one small suggestion. In the section where the implications of the present results to the motion of bubbles, there are no references. The authors make very specific claims but provide no citations. Adding one or two references to classic papers in this subject would be important.*

Response: We have included two references (Refs. 37 & 38) in the final revised manuscript).